# Combat Sports as a Model for Measuring the Effects of Repeated Head Impacts on Autonomic Brain Function: A Brief Report of Pilot Data

Christopher Kirk *,† and Charmaine Childs †

Health Research Institute, Sheffield Hallam University, Sheffield S10 2NA, UK; c.childs@shu.ac.uk
* Correspondence: c.kirk@shu.ac.uk
† These authors contributed equally to this work.

**Abstract:** Automated pupil light reflex (PLR) is a valid indicator of dysfunctional autonomic brain function following traumatic brain injury. PLR's use in identifying disturbed autonomic brain function following repeated head impacts without outwardly visible symptoms has not yet been examined. As a combat sport featuring repeated 'sub-concussive' head impacts, mixed martial arts (MMA) sparring may provide a model to understand such changes. The aim of this pilot study was to explore which, if any, PLR variables are affected by MMA sparring. A cohort of $n = 7$ MMA athletes (age = 24 ± 3 years; mass = 76.5 ± 9 kg; stature = 176.4 ± 8.5 cm) took part in their regular sparring sessions (eight rounds × 3 min: 1 min recovery). PLR of both eyes was measured immediately pre- and post-sparring using a Neuroptic NPi-200. Bayesian paired samples $t$-tests ($BF_{10} \geq 3$) revealed decreased maximum pupil size ($BF_{10} = 3$), decreased minimum pupil size ($BF_{10} = 4$) and reduced PLR latency ($BF_{10} = 3$) post-sparring. Anisocoria was present prior to sparring and increased post-sparring, with both eyes having different minimum and maximum pupil sizes ($BF_{10} = 3$–4) and constriction velocities post-sparring ($BF_{10} = 3$). These pilot data suggest repeated head impacts may cause disturbances to autonomic brain function in the absence of outwardly visible symptoms. These results provide direction for cohort-controlled studies to formally investigate the potential changes observed.

**Keywords:** combat sports; pupillometry; autonomic brain function





## 1. Introduction

Incidence of impact-induced brain trauma has been estimated to be approximately 10 million globally per year, with this considered a conservative assessment [1]. Impacts resulting in head accelerations/decelerations occur during work, leisure and sports related activity with the brain's chronic responses being potentially causative of degenerative brain diseases including Alzheimer's, dementia and chronic traumatic encephalopathy (CTE) [2,3]. Whilst the popular view is that brain trauma is mainly associated with impacts resulting in loss of consciousness (LOC), there is mounting evidence that regular, repeated acceleration/deceleration contra-coup forces to the vault and underlying cerebral tissue may over time increase the risk of brain damage even in the absence of LOC [4–7]. Due to the regularity of such head accelerations/decelerations in sports, the Department of Culture, Media and Sport in the UK recently declared sports-related concussions and sub-concussive events to be a 'public health risk' [8].

Perturbed brain function is the primary symptom of brain trauma with various identification measures being available including balance, eye tracking and memory recall [9,10]. Such methods can, however, be time consuming, subjective and insensitive to changes in autonomic brain function following 'sub-concussive' events without LOC [11–13]. However, if prompt and rapid identification of altered function was available during or immediately following head impacts or accelerations/decelerations, sufferers could be provided with

immediate care. Equally, in sport settings it is thought to be common practice for athletes to continue participation following head impacts due to the absence of visible symptoms of brain injury [14]. Rapid identification of perturbed brain function may allow participants to be protected from further damage resulting from subsequent head impacts that may occur. In each case people with brain trauma may avoid repeated insults that may exacerbate existing incipient neuronal and structural damage [15].

Alterations to the pupillary light response (PLR) is indicative of inhibited autonomic brain function resulting from head impacts [16]. Changes in pupil size has traditionally been determined by a physician who subjectively examines pupil response with a pen torch as 'brisk' or 'sluggish'. The recent development of automated PLR technology, however, provides additional parameters to the assessment of autonomic brain function to allow a reproducible, accurate and quantitative measurement of the eye–brain axis response to light [16,17]. Measured variables include the minimum and maximum pupil size, the velocity and latency of pupil constriction and dilation, and a Neuroptics proprietary variable named 'NPi' which is proposed as a global indicator of PLR. Changes to these variables have been found to be related to the occurrence of brain trauma resulting from both LOC and non-LOC events, including diagnosed concussions [18–20]. Automated PLR has been shown to have greater inter-rater agreement than manual measurement for this purpose [16]. Although there is support for the value of PLR in detection of early clinical deterioration in hospitalised patients with acquired brain injuries following stroke and severe head trauma [21] our collective understanding of the nuances of PLR changes in both clinical and applied settings is in its infancy. There are limited data reporting PLR changes within 72 h of LOC and/or confirmed concussive events [19]. The scarcity of PLR data from repeated non-LOC events is of particular interest given the potential link between such trauma and brain degeneration [4–7]. Unfortunately, study of such events is difficult due to these neurological injuries being outwardly asymptomatic. People who do not demonstrate recognised signs of brain dysfunction are therefore likely to go undiagnosed. The result is an absence of PLR studies from events that would be described as 'sub-concussive', including repeated non-LOC head impacts such as those occurring during combat and contact sports. As such, it is not currently possible to state which, if any, PLR responses would have utility for identifying brain trauma in the acute time frame or in an applied setting such as a place of employment or during a sport or leisure activity.

A potential model for studying the acute effects of a single non-LOC or repeated head accelerations on autonomic brain function is provided by combat sports. Training to compete in boxing, mixed martial arts (MMA) and muay Thai requires participants to experience repeated direct and indirect head accelerations/decelerations from punches, kicks and grappling-based trips, throws and takedowns on an almost daily basis [22,23]. Determining how automated PLR variables respond following combat sports training may provide insight into how this technology may be used to assess brain function post non-LOC impacts. As such, this pilot study aimed to determine which, if any, automated PLR variables are altered immediately following MMA sparring. These results may inform the design, aim and purpose of future cohort-controlled and longitudinal studies into the acute effects of single and/or repeated non-LOC head impacts on PLR measurements. The hypothesis was that PLR variables would be altered by showing either a statistically relevant increase or decrease following MMA sparring. These data may inform future studies into the potential application of this technology in field settings.

## 2. Materials and Methods

To begin examination of the potential of combat sports as a model for understanding automated PLR changes, pilot data were collected from a convenience sample $n = 7$ MMA athletes (age = $24 \pm 3$ years; body mass = $76.5 \pm 9$ kg; stature = $176.4 \pm 8.5$ cm) taking part in their regular sparring session without intervention (8 rounds × 3 min: 1 min recovery). The following inclusion criteria were applied: ≥18 years old; ≥4 official MMA bouts; MMA training age ≥ 2 years. All participants provided written informed consent prior to taking

part in any data collection. PLR of both eyes was measured once from each eye immediately pre- and post-sparring with participants in a seated position using an automated pupilometer (NPi-200, Neuroptics, Irvine, CA, USA). This infrared monocular device takes pupil size measurements automatically from a 3.2 s video recording (30 frames·s$^{-1}$) of the eye following a white light flash of fixed intensity (1000 Lux) and duration (0.8 s). Pre-measurements were taken a minimum of 10 mins after the participants entered the training room to allow for ambient light adaptation [24]. Ambient light range was 30–60 lux throughout the data collection and sparring session (Mini Light and Temp Meter, PEL, Diss, Norfolk, UK). Participants did not leave the training room at any point during the data collection or sparring period. During PLR measurement participants were asked to focus their vision on an 8 cm diameter paper red dot fixed to the wall 130 cm perpendicular from the floor. Participants were seated 200 cm away from the wall on which the red dot was placed. This was done to ensure all participants' pupils were facing in the same direction and focussed on the same point during measurement.

PLR variables collected were as follows: NPi (arbitrary units, AU)—a Neuroptics proprietary variable used to provide an quantitative measure of overall pupil response, with a threshold of <3 considered 'abnormal'; maximum pupil size (mm)—maximum pupil diameter prior to constriction; minimum pupil size (mm)—pupil diameter at peak constriction; absolute amplitude—pupil size change following light stimulus (mm); relative amplitude (%)—percentage of pupil size change following light stimulus; mean constriction velocity (mm·s$^{-1}$)—average velocity of pupil diameter change; maximum constriction velocity (mm·s$^{-1}$)—maximum velocity of pupil diameter change; dilation velocity (mm·s$^{-1}$)—average velocity of pupil returning to resting state following constriction; latency (s)—time to constriction onset following light stimulus [25].

Statistically relevant differences between pre and post measurements for all variables were determined by Bayes factors (BF$_{10}$) $\geq$ 3 from paired samples *t* tests using a JZS Cauchy prior = 0.707 with location parameter = 0. The following thresholds were used for each BF$_{10}$: 1–2.9 = anecdotal; 3–9.9 = moderate; 10–29.9 = strong; 30–99.9 = very strong; $\geq$100 = decisive. Due to default priors being used, BF$_{10}$ robustness checks were performed. Where a result was found to cross a threshold, both thresholds are reported [26]. For brevity, *p* values are not reported, but any result found to support a hypothesis (BF$_{10}$ $\geq$ 3) was also found to have acceptably low probability of type 1 error (*p* < 0.05). Any result found to have BF$_{10}$ < 3 was deemed inconclusive. Effect sizes were calculated via Cohen's d using the standard deviation of the mean scores as the denominator. The following thresholds were used for Cohen's d: small $\geq$ 0.2; moderate $\geq$ 0.6; large $\geq$ 1.2; very large $\geq$ 2. All statistical analyses were completed using JASP 0.16.30. This pilot study was conducted in accordance with the Declaration of Helsinki and approved by the ethics committee of Sheffield Hallam University (ER36006171, 3 August 2022).

## 3. Results

Whilst all participants experienced multiple direct and indirect head accelerations/decelerations from punches, kicks and throws during the session, PLR variables from the mean of both eyes (Table 1) showed increased NPi (BF$_{10}$ = 4; d = 1.2), decreased maximum pupil size (BF$_{10}$ = 3; d = 1), decreased minimum pupil size (BF$_{10}$ = 4; d = 1.2) and reduced PLR latency (BF$_{10}$ = 3; d = 1.1). When comparing the right eye (R) to the left eye (L) (Table 2), anisocoria was present prior to sparring and then increased post-sparring. Each eye had different minimum pupil sizes both pre- (BF$_{10}$ = 4; d = 1) and post-sparring (BF$_{10}$ = 3; d = 1.1). Each eye also had different maximum pupil sizes (BF$_{10}$ = 4; d = 1.2) and constriction velocities (BF$_{10}$ = 3; d = 1.1) post-sparring.

**Table 1.** PLR variables averaged between both eyes pre- and post-MMA sparring bouts.

|  | Pre | Post |
|---|---|---|
| NPi (AU) * | 4.1 ± 0.3 | 4.2 ± 0.3 |
| Maximum pupil size (mm) * | 5.8 ± 1.1 | 5.4 ± 1 |
| Minimum pupil size (mm) * | 3.8 ± 0.9 | 3.4 ± 0.7 |
| Mean constriction velocity (mm·s$^{-1}$) | 2.9 ± 0.4 | 3.2 ± 0.6 |
| Maximum constriction velocity mm·s$^{-1}$) | 4.5 ± 0.7 | 5.1 ± 0.9 |
| Dilation velocity (mm·s$^{-1}$) | 1.2 ± 0.2 | 1.4 ± 0.2 |
| Latency (s) * | 0.23 ± 0.02 | 0.20 ± 0.01 |
| Absolute amplitude (mm) | 1.96 ± 0.28 | 1.93 ± 0.39 |
| Relative amplitude (%) | 34.3 ± 3.6 | 36 ± 2.6 |

* Statistically relevant change Pre-Post (BF$_{10}$ ≥ 3); data show mean ± SD.

**Table 2.** PLR variables comparing left eye to right eye pre- and post-MMA sparring bouts.

|  | Pre | | Post | |
|---|---|---|---|---|
|  | Left Eye | Right Eye | Left Eye | Right Eye |
| NPi (AU) | 4.07 ± 0.3 | 4 ± 0.4 | 4.2 ± 0.3 | 4.2 ± 0.3 |
| Maximum pupil size (mm) | 5.7 ± 1.1 | 5.9 ± 1 | 5.2 ± 1 | 5.6 ± 1 * |
| Minimum pupil size (mm) | 3.7 ± 0.9 | 3.9 ± 0.9 * | 3.3 ± 0.7 | 3.5 ± 0.7 * |
| Mean constriction velocity (mm·s$^{-1}$) | 2.9 ± 0.4 | 3 ± 0.3 | 2.9 ± 0.8 | 3.5 ± 0.6 |
| Maximum constriction velocity (mm·s$^{-1}$) | 4.3 ± 0.7 | 4.6 ± 0.7 | 4.8 ± 1.1 | 5.4 ± 0.8 * |
| Dilation velocity (mm·s$^{-1}$) | 1.1 ± 0.3 | 1.3 ± 0.2 | 1.4 ± 0.2 | 1.4 ± 0.2 |
| Latency (s) | 0.22 ± 0.03 | 0.24 ± 0.04 | 0.20 ± 0.02 | 0.21 ± 0.02 |
| Absolute amplitude (mm) | 1.93 ± 0.36 | 2 ± 0.25 | 1.85 ± 0.47 | 2.02 ± 0.34 |
| Relative amplitude (%) | 34.4 ± 3.6 | 34.1 ± 4.3 | 35.6 ± 3.8 | 36.4 ± 2.2 |

* Statistically relevant difference between eyes (BF$_{10}$ ≥ 3); data show mean ± SD.

## 4. Discussion

The aim of this pilot study was to determine which, if any, automated PLR variables are altered immediately following MMA sparring. The hypothesis was that PLR variables would be altered by showing either a statistically relevant increase or decrease following MMA sparring. It was found that NPi increased, whilst maximum pupil size and minimum pupil size decreased immediately post-sparring. PLR was also found to have a shorter latency period. Anisocoria was revealed to be present in this cohort prior to sparring and this was exacerbated by sparring.

PLR is controlled by the autonomic nervous system. Pupil constriction is caused by increased parasympathetic activity with concurrently reduced sympathetic activity. Pupil dilation results from the reverse of this, with parasympathetic activity being inhibited and sympathetic activity increased [27]. The dual response of these systems to a light stimulus occurs along a complex network of neuronal structures linking the brain and the sphincter muscles of the eye [28]. Accordingly, disturbed function of any component or connection in this network would have an effect on the measured PLR [17]. The repeated coup-contrecoup acceleration and deceleration of the head experienced during combat sports performance and training may cause such disturbances. MMA athletes have been found to experience an average of 15.7 head impacts with acceleration >10 g in sparring sessions. These impacts result in peak angular acceleration = 2,149 ± 14,285 rad·s$^2$ and peak linear acceleration = 32 ± 17.2 g, with the energy exerted on the brain = 5 ± 3.1 kW during the greatest of these [29]. These forces may cause Tresca shear stress of the corpus callosum which is related to incidences of concussion in competitive bouts [30]. Even in the absence of diagnosable concussion (using subjective tools such as the SCAT5), head impacts resulting in accelerations less than the aforementioned values have been related to PLR changes measured over several months in 14–19 year old athletes [31]. Though no participant in this cohort displayed outward symptoms of concussion, all were observed to receive repeated head impacts. 'Sub-concussive' events such as these have previously

resulted in acute autonomic brain function decrements such as increased corticomotor inhibition, altered motor unit recruitment and decreased memory following combat sports sparring [6]. It may therefore be expected that these decrements may also manifest as disturbed PLR following sparring despite an absence of LOC. These pilot data provide potential support for this, revealing that there are a range of PLR changes following combat sports sparring that require further examination.

NPi showed a statistically relevant increase following sparring. This may indicate that no neurological insult has occurred given the manufacturer recommendation of NPi < 3 being 'abnormal'. Norm values for NPi amongst neurocritical patients have, however, been reported as $4.2 \pm 0.8$ AU, with even those deemed as being 'severe' according to the Glasgow coma scale (GCS) = $4.1 \pm 0.9$ AU, similar to the NPi of the current cohort [20]. This suggests that an NPi $\leq 3$ may not be a sensitive marker of acute brain dysfunction, especially in the absence of concussion symptoms.

The expected pupillary response to increased exercise intensity and emotional arousal is increased dilation rather increased constriction [32–34]. MMA sparring is by nature a high intensity activity [35]. Despite this, pupil constriction increased with reduced latency in our cohort following sparring. This shows the participant's pupils responded more rapidly to light stimulus and were more constricted both prior to this stimulus and after. Increased pupil constriction has been observed in mild traumatic brain injury (mTBI) patients compared to healthy controls [36] and also in comparison to normative values [20]. The participants observed in the current study were found to have pupil diameters below the norm values for their age range both pre- and post-sparring [37]. Decreased maximal pupil diameters have also been recorded in mixed-sex mTBI patients 1 year after suffering a head impact [36]. MMA participation is associated with reduced brain white matter volume and processing speeds [4,38], with these changes potentially being exacerbating over time with repeated head impacts [5]. Further, more severe decrements may be related to an earlier age of first exposure to combat sport participation [39]. Though speculative at this time, it may be the case that changes in pupil diameter measures may be related to chronic neural maladaptation.

Dilation and constriction velocities all increased following sparring, though these were not found to be statistically relevant. Whilst these variables have been found to be faster in concussed children [40], they appear to become slower in concussed adults [27] with further decrements in velocity for each level of GCS severity [18]. As such, any potential acute brain dysfunction experienced by these participants appears to have been 'minor' according to our current understanding of PLR changes following trauma. Similar to this cohort's pupil diameters, however, both dilation and constriction velocities were found to be below the norm values expected for healthy, age matched participants [37]. These data taken together may demonstrate autonomic disturbances that may already be present in this population, and that they may be made acutely worse by sparring. Reflecting previous work [41], these data may indicate that acute and chronic alterations to PLR variables may differ between 'concussive' and 'sub-concussive' events. This may also be the case for changes over time, where the pattern of PLR changes and 'recovery' may be important for determining neurological effect [31,42].

There were no statistical changes to either relative or absolute amplitude following MMA sparring. This supports the notion of the observed changes being due to disturbance or alterations in the midbrain, rather than local eye functions [17,28]. This may be important for future studies relating PLR variables to measurable changes brain structure or function. It is important to note, however, that whilst these variables did not show statistically relevant differences, the pre–post changes in these measurements may show clinically meaningful changes [43]. Understanding this would require measurements of pupil stability prior to sparring or physical activity to quantify the noise inherent in a dynamic system. This would also allow any changes to be appreciated within context. Future studies should, therefore, include multiple pre-sparring measures under controlled conditions to allow pupil stability and clinically meaningful changes to be determined.

The worsening of anisocoria reported here may also indicate acute autonomic brain function disturbance. PLR differences between eyes has been suggested to not have a mechanistic relation to brain dysfunction as both hemispheres of the brain are presumed to be equally affected by impact so should effect both eyes to the same extent [17]. Despite this, anisocoria has been shown to be greater in mTBI patients than uninjured people [44], and also to be indicative of more severe GCS classification and reduced recovery [45]. The cohort in the present study displayed post-sparring differences in maximum pupil size between eyes that breached the suggested 'abnormal' threshold of 0.3 mm [46]. This is in addition to large differences between eyes in minimum pupil size and constriction velocity post-sparring. These data may show acute autonomic brain dysfunction caused by repeated non-LOC head impacts. There is an absence of data, however, regarding the effects of high intensity exercise on anisocoria. There is also a sparsity of studies examining such occurrence over time. As such, whilst these results could suggest a neurological effect that can be measured immediately post-sparring, it is not currently possible to rule out this being a result of exercise induced acute fatigue or physiological arousal.

## 5. Limitations

The effects of exercise-induced arousal and fatigue cannot be entirely ruled out from the reported data. Neither can these results be linked directly to the frequency or magnitude of the head accelerations/decelerations experienced due to the lack of measures such as instrumented gumshields [29]. The reported changes were also measured immediately post-sparring, with no data currently available regarding the 'recovery' or 'decay' of these variables. Equally, the use of only one PLR measurement per eye prior to sparring meant that pupil stability of the cohort remains unknown. Future studies should therefore include a series of repeat measures prior to sparring and over several hours and days post-sparring to determine if/how PLR variables differ and/or resolve once any potential arousal or fatigue has subsided, and what the decay time of this might be. Repeat measures may also reveal if or how PLR variables respond at different time points following repeated, non-LOC head accelerations/decelerations [41]. These studies should include combat sport and non-combat sport populations in cohort-matched controls to allow a fuller contextual evaluation of any identifiable patterns. It would also be pertinent to combine these measures with data quantifying the frequency and magnitude of training and competition-based head accelerations to determine potential exposure thresholds related to PLR variable responses.

This study includes a relatively small sample size due to the low number of experienced combat sports participants fitting the inclusion criteria available. Bayesian analyses were therefore used to account for the small sample size with the Bayes factor itself representing the strength or otherwise of the evidence [47]. This contrasts the comparison of observed data to hypothetical repeat trials as is the case in frequentist analyses [48]. As such, 'power' in a frequentist sense is not applicable to Bayesian methods, meaning the presented analysis results are not dependent on sample size [47].

## 6. Conclusions

These preliminary pilot data suggest that there may be some acute changes in autonomic brain function following non-LOC, repeated head accelerations/decelerations brought about by combat sport sparring. These appear to manifest as increased pupil constriction, reduced PLR latency and increased anisocoria in terms of pupil size and constriction velocity as measured by automated PLR technology. The results appear to be different to those reported from mTBI patients and sport participants after experiencing diagnosable concussive events.

In conclusion, using combat sports as a model for examining PLR responses to head impacts may provide a greater understanding of autonomic brain function alterations in response to acute, 'sub-concussive' and non-LOC related brain trauma. This may allow the development of thresholds and 'early warning' markers for use in sport, clinical and work settings via a rapid (<1 min) measurement. This information may prove invaluable for

the reduction of long-term brain degenerative diseases linked to repeated non-LOC head impacts. These preliminary results may now inform the aims, questions and designs of future cohort-controlled studies to investigate the application and utility of these variables in understanding the effects of 'sub-concussive' head impacts on health and performance.

## 7. Declarations

The authors have no relevant financial or non-financial conflicts of interest to report. There was no funding to declare for this work. For the purpose of open access, the author has applied a Creative Commons Attribution (CC BY) licence to any Author Accepted Manuscript version arising from this submission.

**Author Contributions:** Conceptualization, C.K. and C.C.; methodology, C.K. and C.C.; software, C.K.; formal analysis, C.K.; investigation, C.K. and C.C.; resources, C.C.; writing—original draft preparation, C.K.; writing—review and editing, C.K. and C.C.; project administration, C.K. All authors have read and agreed to the published version of the manuscript.

**Funding:** This research received no external funding.

**Institutional Review Board Statement:** This pilot study was conducted in accordance with the Declaration of Helsinki and approved by the ethics committee of Sheffield Hallam University (ER36006171, 3 August 2022).

**Informed Consent Statement:** Informed consent was obtained from all participants involved in the study.

**Data Availability Statement:** Please contact the authors to discuss data availability and sharing.

**Conflicts of Interest:** There are no conflicts of interest to report for this study.

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
