# Peer review of "Combat Sports as a Model for Measuring the Effects of Repeated Head Impacts on Autonomic Brain Function: A Brief Report of Pilot Data"

_2411-5150, 2023_

Round 1

Reviewer 1 Report

The findings are interesting! I have a few major concerns that need to be addressed:

1.       What was the pupillometry light protocol applied in the study? Without the light stimulus details, it is difficult to interpret the findings. Please provide the intensity, wavelength and duration of the light pulse. How was maximum pupil size determined?

2.       What was the room illumination before and after the MMA session? Was it the same? If not, the differences observed before and after the MMA session could be just due to change in room illumination. This is critical.

3.       Fatigue can make the pupil smaller. Is this what the athletes are showing after the MMA session?

4.       As per the published ‘Standards in Pupillography’, the minimum pupil size needs to be normalised to the maximum pupil size to determine the pupil light response amplitude. Please do so and evaluate whether this alters your findings and interpretation. Please revise the manuscript accordingly.

5.       Given the amount of data presented, the introduction and discussion are too long. Please make them as succinct as possible by avoiding speculations.

Author Response

The findings are interesting! I have a few major concerns that need to be addressed:

1. What was the pupillometry light protocol applied in the study? Without the light stimulus details, it is difficult to interpret the findings. Please provide the intensity, wavelength and duration of the light pulse. How was maximum pupil size determined?

Response: Thank you for highlighting this oversight on our part. The methods section has now been amended to include the following details:

“This infrared monocular device takes pupil size measurements automatically from a 3.2 s video recording (30 frames∙s-1) of the eye following a white light flash of fixed intensity (1,000 Lux) and duration (0.8 s).”

2. What was the room illumination before and after the MMA session? Was it the same? If not, the differences observed before and after the MMA session could be just due to change in room illumination. This is critical.

Response: Thank you for highlighting this oversight on our part. The methods section has now been amended to include the following details:

“Pre measurements were taken a minimum of 10 minutes after the participants entered the training room to allow for ambient light adaptation[24]. Ambient light range was 30-50 Lux throughout the data collection and sparring session (Mini Light and Temp Meter, PEL, Norfolk). Participants did not leave the training room at any point during data collection or sparring period.”

3. Fatigue can make the pupil smaller. Is this what the athletes are showing after the MMA session?

Response: To our knowledge constriction occurs under fatigue caused by sleep deprivation or cognitive tasks longer than ~40 mins. This cohort was not sleep deprived or engaged in cognitive tasks of the types used in such research, so the fatigue which brings about such constriction is unlikely to be present in this study. In addition, physical tasks of similar duration and intensity are related to increased dilation. Using the Kuwamizu et al. 2022 study as comparison, it can be seen that exercise incorporating the intensity range of MMA sparring (Kirk et al., 2021) results in pupil dilation, not constriction. As such, we do not believe that these changes are caused by fatigue, but we do mention the potential for this in the Limitations section, and suggest future studies to control for this in the Discussion section.

4. As per the published ‘Standards in Pupillography’, the minimum pupil size needs to be normalised to the maximum pupil size to determine the pupil light response amplitude. Please do so and evaluate whether this alters your findings and interpretation. Please revise the manuscript accordingly.

Response: Relative amplitude was reported in the originally submitted manuscript via ‘Pupil size change following light stimulus (%)’. We have now renamed this variable as ‘Relative amplitude (%)’. To ensure full reporting of data and to fit in with the recommendations of Kelbsch et al., (2019), we have now also added the variable ‘Absolute amplitude (mm)’ for each eye and the average between eyes with the required statistical analyses. There is no change to the findings or interpretations as a result of this additional variable.

5. Given the amount of data presented, the introduction and discussion are too long. Please make them as succinct as possible by avoiding speculations.

Response: Respectfully, the stated aim of this report is to report pilot data from a study designed to provide an initial exploration of the pupil response to combat sports participation and suggest directions for future studies. As such, we would argue that speculation of what may have caused these results is required in order to point researchers towards future hypotheses and studies. Additionally, we have been requested to provide an article of at least 3,000 words by the editing office so are unable to reduce the length of these sections.

Reviewer 2 Report

Thanks for the opportunity to review this work.  The paper presents the interesting idea of evaluating the pupil light response (PLR) or pupil light reflex prior to and after a small sample of participants completed a regular combat sports sparring session.  Results of this pilot study demonstrated a pre- to post- change in the minimum and maximum PLR and a change in the NPi (i.e., proprietary measure of PLR provided by the equipment manufacturer).  Based on these findings, the authors report the PLR provides a possible measure for evaluating subconcussive or asymptomatic concussive events.

As an oculomotor control researcher with interest in pupillometry and concussion, I think this is an interesting study; however, the study requires a control condition (i.e., pre- to post-evaluation of PLR without combat training) and also a session involving a non-combat exercise of equivalent intensity and duration.  I understand from the Discussion the authors suggested that this finding cannot be attributed to factors such as exercise intensity or cognitive load; however, the papers the authors cite to support their justification are not directly relevant to the current study.  In particular, Kuwamizu et al. (2022: J Physiol Sci) evaluated pupil size during a graded exercise test.  The authors of that study attempted to continuously measure pupil size (and not the PLR) and this was measured during exercise.  Indeed, there are many variables that impact the PLR during, concurrently and after exercise and during, concurrently and after a cognitive task performance (for excellent review see Mathot 2018: J Cogn; see also Wang and Munoz 2015: Curr Opin Neurobiol).  Hence, it is entirely necessary to compare results to a relevant control condition(s).

Introduction needs detail on the nature and neural underpinnings of the PLR.  Also, has any other work employed pupillometry to evaluate concussive or subconcussive impairments? 

It was not clear to me what the authors were measuring.  Was onset of the “red dot” (i.e., the stimulus) the time at which the PLR was measured.  A schematic showing onset of the stimulus and an exemplar PLR is warranted.

How was the sample size derived?

What was the luminance of the “red dot”?  Further, in the visual sciences the size of a target or stimulus is reported in degrees of visual angle.

What was the luminance of the testing environment and was it held constant between- and within-participants?  How long before and after sparring did the assessment take place and did you inquire with participants as to whether their perception of the “red dot” brightness changed from pre- to post-assessment intervals (i.e., perception of brightness has a reliable impact on the PLR). 

How many times was the PLR assessed at pre- and post- time points? Was it a single measurement timepoint?

Author Response

As an oculomotor control researcher with interest in pupillometry and concussion, I think this is an interesting study; however, the study requires a control condition (i.e., pre- to post-evaluation of PLR without combat training) and also a session involving a non-combat exercise of equivalent intensity and duration.  I understand from the Discussion the authors suggested that this finding cannot be attributed to factors such as exercise intensity or cognitive load; however, the papers the authors cite to support their justification are not directly relevant to the current study.  In particular, Kuwamizu et al. (2022: J Physiol Sci) evaluated pupil size during a graded exercise test.  The authors of that study attempted to continuously measure pupil size (and not the PLR) and this was measured during exercise.  Indeed, there are many variables that impact the PLR during, concurrently and after exercise and during, concurrently and after a cognitive task performance (for excellent review see Mathot 2018: J Cogn; see also Wang and Munoz 2015: Curr Opin Neurobiol).  Hence, it is entirely necessary to compare results to a relevant control condition(s).

Response: We agree that a control group would be required to confirm any specific changes in a study designed to answer any specific hypotheses or provide any firm conclusions. This study was, however, always designed as a pilot to begin the exploration of which variables should be the subject of specific investigations, and which types of controlled study design(s) should be targeted for resource allocation. As such, it was decided to not include a control until it was known what kind(s) of control group(s) would be needed for the condition(s) and variable(s) that may be of interest. To ensure the readers are fully aware of these intentions the following amendment has been added to the aims in the final paragraph of the introduction:

“These results may inform the design, aim and purpose of future cohort controlled and longitudinal studies into the acute effects of single and/or repeated non-LOC head impacts on PLR measurements.”

Regarding the use of Kuwamizu et al as a comparison study, we respectfully argue that this paper is directly relevant to our results. In this section we are specifically discussing the pupil size, not the overall PLR, so direct comparison to the changes in pupil size reported by Kuwamizu et al. is warranted as this reference is used to show the expected pupil size change during and following exercise is dilation, not the constriction we have reported here. Our cohort displaying the exact opposite of what is expected in terms of pupil size change is therefore relevant. Finally, the exercise being undertaken in Kuwamizu et al. is graded, but does incorporate the intensity and physiological arousal range expected of MMA sparring (Kirk et al., 2021; Petersen & Lyndsay, 2020) providing a relevant comparison. Overall, the point being made in this section is that constriction is not the expected response to exercise, making these references (Kuwamizu; Pan et al.,; Bradely et al.) and discussion entirely relevant.

We do, however, agree that reference to ‘cognitive involvement’ is misplaced here, and we have now removed this statement from this section to ensure the discussion focusses on physiological arousal, as this can be directly related to MMA sparring from the previously referenced studies, whilst cognitive involvement we accept cannot be related at this stage. Thank you for the references regarding PLR and cognitive tasks, we will make sure to use these for future cohort-controlled designs.

Introduction needs detail on the nature and neural underpinnings of the PLR.  Also, has any other work employed pupillometry to evaluate concussive or subconcussive impairments? 

Response: Thank you for this suggestion. We do not believe, however, that this particular brief report is the appropriate place to include a detailed discussion of the physiological underpinnings of PLR given the shorter word count and purpose of a brief report. We have, however, included several references to published works that provide this detail and information, and we included the following statement in lines 145-153 of the originally submitted manuscript that highlights the neuronal nature of this response, with this section now appearing on lines 161-168 of the resubmitted manuscript:

“PLR is controlled by the autonomic nervous system. Pupil constriction is caused by increased parasympathetic activity with concurrently reduced sympathetic activity. Pupil dilation results from the reverse of this, with parasympathetic activity being inhibited and sympathetic activity increased [26]. The dual response of these systems to a light stimulus occurs along a complex network of neuronal structures linking the brain and the sphincter muscles of the eye [27]. Accordingly, disturbed function of any component or connection in this network would have an effect on the measured PLR [17]. The repeated coup-contrecoup acceleration and deceleration of the head experienced during combat sports performance and training may cause such disturbances.”

Regarding the use of PLR for concussion and sub-concussive work, the introduction provides references 16-21 in the original manuscript (they remain as references 16-21 in the resubmitted manuscript) pointing readers to previous studies showing the use of this measure for concussions. The introduction in the originally submitted manuscript also included the following statement highlighting the lack of work using PLR for sub-concussions, hence the completion of this study in the first instance:

“There are limited data reporting PLR changes within 72 hours of LOC and/or confirmed concussive events [19]. The scarcity of PLR data from repeated non-LOC events is of particular interest given the potential link between such trauma and brain degeneration [4–7]. Unfortunately, study of such events is difficult due to these neurological injuries being outwardly asymptomatic. People not demonstrating recognised signs of brain dysfunction are therefore likely to go undiagnosed. As such, it is not currently possible to state which PLR responses would have utility for identifying brain trauma in the acute time frame or in an applied setting such as a place of employment or during a sport or leisure activity.”

It was not clear to me what the authors were measuring.  Was onset of the “red dot” (i.e., the stimulus) the time at which the PLR was measured.  A schematic showing onset of the stimulus and an exemplar PLR is warranted.

Response: We apologise for the misunderstanding that our wording in this section of the methods has caused. The red dot was not the stimulus, the stimulus is provided by the NPi device itself. The red dot is a paper based focal point there to merely ensure that all participant’s eyes were facing the same direction and focused on the same area. Each of these points have now been clarified with the following amendments to the methods:

“All participants provided written informed consent prior to taking part in any data collection. PLR of both eyes was measured once from each eye immediately pre and post sparring with participants in a seated position using an automated pupilometer (NPi-200, Neuroptics, USA). This infrared monocular device takes pupil size measurements automatically from a 3.2 s video recording (30 frames∙s-1) of the eye following a white light flash of fixed intensity (1,000 Lux) and duration (0.8 s). Pre measurements were taken a minimum of 10 minutes after the participants entered the training room to allow for ambient light adaptation [24]. Ambient light range was 30-60 Lux throughout the data collection and sparring session (Mini Light and Temp Meter, PEL, UK). Participants did not leave the training room at any point during data collection or sparring period. During PLR measurement participants were asked to focus their vision on an 8 cm diameter paper red dot fixed to the wall 130 cm perpendicular from the floor. Participants were seated 200 cm away from the wall on which the red dot was placed. This was done to ensure all participant’s pupils were facing in the same direction and focussed on the same point during measurement.”

How was the sample size derived?

Response: This sample was a convenience sample based on the following inclusion criteria, which we have now been added to the methods section:

“The following inclusion criteria were applied: > 18 years; ≥ 4 official MMA bouts; MMA training age ≥ 2 years.”

Whilst this is a small sample, it is in keeping with the nature of this study as a pilot that was not intended to be a large cohort study. The use of Bayesian analyses reduces the requirement of a large sample, particularly for exploratory work. We have now discussed this within the limitations paragraph with the following statement:

“This study includes a relatively small sample size due to the low number of experienced combat sports participants fitting the inclusion criteria available. Bayesian analyses were therefore used to account for the small sample size with the Bayes factor itself representing the strength or otherwise of the evidence [46]. This contrasts the comparison of observed data to hypothetical repeat trials as is the case in frequentist analyses [47]. As such, ‘power’ in a frequentist sense is not applicable to Bayesian methods, meaning the presented analysis results are not dependent on sample size [46].”   

What was the luminance of the “red dot”?  Further, in the visual sciences the size of a target or stimulus is reported in degrees of visual angle.

 Response: Please see previous clarification of the purpose of the red dot.

What was the luminance of the testing environment and was it held constant between- and within-participants?  How long before and after sparring did the assessment take place and did you inquire with participants as to whether their perception of the “red dot” brightness changed from pre- to post-assessment intervals (i.e., perception of brightness has a reliable impact on the PLR). 

Response: Thank you for highlighting our oversight regarding the ambient light conditions. The following statement has now been added to the methods:

 “Pre measurements were taken a minimum of 10 minutes after the participants entered the training room to allow for ambient light adaptation[24]. Ambient light range was 30-60 Lux throughout the data collection and sparring session (Mini Light and Temp Meter, PEL, Norfolk). Participants did not leave the training room at any point during data collection or sparring period.”

Regarding the timing of the measurements, the originally submitted manuscript stated “PLR of both eyes was measured immediately pre and post sparring” on line 89-90, with this now appearing on lines 95-96 of the resubmitted manuscript. Please also see previous clarification of the purpose of the red dot.

How many times was the PLR assessed at pre- and post- time points? Was it a single measurement timepoint?

Response: Thank you for highlighting this oversight on our part. PLR was measured once from each eye immediately pre and post, with this now being clarified in the methods section.

Reviewer 3 Report

The purpose of the current brief report was to if PLR (pupillary light response) are changed following mixed martial arts (MMA) type sparring sessions. Based on prior research the authors hypothesized that some PLR variables would increase, whereas others would decrease following the sparring session in which participants would experience head impacts.  A total of 7 MMA athetes completed the study and completed one session of sparing that involved 8 rounds that were each 3 minutes long. Automated PLR technology was used to measure or derive 8 PLR variables. In addition, these variables were compared between the two eyes. One of the variables was termed NPi (Neuroptics proprietary variable), which is a global index of PLR.

 The main findings were: 1) four of the variables were significantly different post sparring; 2) NPi increased; 3) maximum pupil size decreased; 4) minimum pupil size decreased; 5) there was reduced PLR latency; 6) anisocoria was present before sparring and increase after sparring; and 7) each eye had different minimum pupil sizes pre and post sparring and different maximum pupil sizes and constriction velocities post sparing.

 Overall, this is an interesting study that has several strengths such as being on a very important topic, athletes were used, and the use of automated PLR technology. The paper was also very well-written, had very few typos or grammatical mistakes, can be built upon with future research, and should be of interest to many readers of the journal. The paper significantly adds to the literature. Thus, I think the paper should be published. I only have several minor comments that should be addressed.

 1.      I know it is a preliminary study and a brief report but the sample size is kind of low, perhaps the authors should add a sentence or two about why this is not an issue.

2.      Relatedly, I think in the 2nd to last paragraph the author should provide some limitations of the study. They sort of infer these in the paragraph but there should probably be more. Such as the sample size and maybe that the actual number of head impacts (and the forces of each) were not measured with some sort of device such as an instrumented mouthpiece etc.

3.      The rest of my comments are just some typos

a.       Lines 92 and 93 I think there should be a space before “cm” in all 3 cases.

b.      Line 100, I think “Dilation” should not be capital as other variables weren’t capitalized.

c.       Line 137, it appears there is no space between the period and the word “Discussion” in the heading.

Author Response

Overall, this is an interesting study that has several strengths such as being on a very important topic, athletes were used, and the use of automated PLR technology. The paper was also very well-written, had very few typos or grammatical mistakes, can be built upon with future research, and should be of interest to many readers of the journal. The paper significantly adds to the literature. Thus, I think the paper should be published. I only have several minor comments that should be addressed.

I know it is a preliminary study and a brief report but the sample size is kind of low, perhaps the authors should add a sentence or two about why this is not an issue.

Response: Whilst this is a small sample, it is in keeping with the nature of this study as a pilot that is not intended to be a large cohort study. The use of Bayesian analyses reduces the requirement of a large sample, particularly for exploratory work. We  are aware that many readers are not fully aware of the strength of Bayesian analyses in this manner, so we have now discussed this within the limitations paragraph with the following statement:

“This study includes a relatively small sample size due to the low number of experienced combat sports participants fitting the inclusion criteria available. Bayesian analyses were therefore used to account for the small sample size with the Bayes factor itself representing the strength or otherwise of the evidence [46]. This contrasts the comparison of observed data to hypothetical repeat trials as is the case in frequentist analyses [47]. As such, ‘power’ in a frequentist sense is not applicable to Bayesian methods, meaning the presented analysis results are not dependent on sample size [46].   

Relatedly, I think in the 2nd to last paragraph the author should provide some limitations of the study. They sort of infer these in the paragraph but there should probably be more. Such as the sample size and maybe that the actual number of head impacts (and the forces of each) were not measured with some sort of device such as an instrumented mouthpiece etc.

Response: We have now reorganised the final third of the discussion to include separate, headed sections for Limitations and Conclusions making more specific reference to each of the stated limitations.

The rest of my comments are just some typos

Lines 92 and 93 I think there should be a space before “cm” in all 3 cases.

Line 100, I think “Dilation” should not be capital as other variables weren’t capitalized.

Line 137, it appears there is no space between the period and the word “Discussion” in the heading.

Response: Thank you for highlighting these errors for us, each has now been corrected.

Round 2

Reviewer 1 Report

Thank you for addressing my comments/questions.

The relative amplitude (%) metric shows no change post-training. This indicates that the training does not affect the local eye functions and therefore the post-training change in the maximum and minimum pupil size is due to alterations in the activity of higher brain centres. I suggest the authors to include such interpretation of their data in the discussion section, which will be informative to future studies to understand the mechanism of the change in pupil response with minor head insults.

With the new pupil metrics analysed, I do not agree with the conclusion statement that the pupil constriction increased post-training because the pupil constriction amplitude (relative amplitude) remained stable. The pupil constriction velocity indeed increased and I would like to encourage the authors to discuss the possible reasons for this increase. Does this imply increased muscular activity/alertness/something else post-training?

The abstract section needs to focus on the original findings and interpretations instead of literature review and hypotheses.

Author Response

The relative amplitude (%) metric shows no change post-training. This indicates that the training does not affect the local eye functions and therefore the post-training change in the maximum and minimum pupil size is due to alterations in the activity of higher brain centres. I suggest the authors to include such interpretation of their data in the discussion section, which will be informative to future studies to understand the mechanism of the change in pupil response with minor head insults.

Thank you for highlighting this, and we agree that this is an important point to make for readers to follow up on in future studies. We have now added the following statement to lines 223-226:

“There were no statistical changes to either relative or absolute amplitude following MMA sparring. This support the observed changes being due to disturbance or alterations in the midbrain, rather than local eye functions [17,28]. This may be important for future studies relating PLR variables to measurable changes brain structure or function.”

With the new pupil metrics analysed, I do not agree with the conclusion statement that the pupil constriction increased post-training because the pupil constriction amplitude (relative amplitude) remained stable. The pupil constriction velocity indeed increased and I would like to encourage the authors to discuss the possible reasons for this increase. Does this imply increased muscular activity/alertness/something else post-training?

We respectfully disagree on these points. We argue that pupil constriction did increase as both the minimum and maximum pupil diameter of both pupils decreased post sparring, showing that the pupils are more constricted following sparring. In addition, constriction velocity did not show a statistically relevant increase, just a difference between eyes post sparring (which has been discussed in the section about anisocoria). A case could be made that whilst relative amplitude did not show a statistically relevant change, it did not remain ‘stable’ (34% v 36%), and also that whilst constriction velocity did not show a statistical change it also did not remain ‘stable’ (mean CV = 2.9 v 3.2; max CV = 4.5 v 5.1). If relative amplitude can be said to not change in lieu of a statistical change, however, then we must also conclude that constriction velocity did not change. As such, we do not wish to make too much of this within the discussion, however, we have added the following statement to lines 223-233 mentioning this point and suggesting it as a potential area for researchers to be aware of in future studies:

“There were no statistical changes to either relative or absolute amplitude following MMA sparring. This support the observed changes being due to disturbance or alterations in the midbrain, rather than local eye functions [17,28]. This may be important for future studies relating PLR variables to measurable changes brain structure or function. It is important to note, however, that whilst these variables did not show statistically relevant differences, the pre-post changes in these measurements may show clinically meaningful changes [43]. Understanding this would require measurements of pupil stability prior to sparring or physical activity to quantify the noise inherent in a dynamic system. This would also allow any changes to be appreciated within context. Future studies should, therefore, include multiple pre-sparring measures under controlled conditions to allow pupil stability and clinically meaningful changes to be determined. “     

The abstract section needs to focus on the original findings and interpretations instead of literature review and hypotheses.

We agree, and have now re-written the abstract accordingly.

Reviewer 2 Report

The paper is styled as a pilot investigation which is fine; however, it does not preclude that a control (between- or within-participants) condition should be employed.  Indeed, the measurement is simple and contextualizing the conclusions as being specific to a combative sport participation merits a control condition.

The authors addressed my comment about how many times pupil size was measured.  Given that it was measured once/eye pre- and post is there any documentation of the stability of pupil size over repeated trials.  In other words, how stable is pupil size at pre-participation?

Author Response

The paper is styled as a pilot investigation which is fine; however, it does not preclude that a control (between- or within-participants) condition should be employed.  Indeed, the measurement is simple and contextualizing the conclusions as being specific to a combative sport participation merits a control condition.

We agree that a control group is required for making firm conclusions regarding any effect. That is why we are now in the recruitment stage of an exercise matched controlled cohort study including multiple pre sparring measures and repeat post sparring measures over several hours and days. This is as a direct result of these pilot data directly informing our choices for this study. Equally, given the 5 days provided to respond to these comments by the journal, it would not be possible to recruit, run and analyse a control group in addition to re-writing large section of this paper. We therefore will not be adding a control group to the current paper as we believe the aim of this pilot study did not warrant it given our previously stated reasons for not including one during the study design. We have, however, added the following statement to conclusions section on lines 289-292 to ensure readers are aware that these data are only to inform future studies and are not presented as being decisive on their own:

“These preliminary results may now inform the aims, questions and designs of future cohort-controlled studies to investigate the application and utility of these variables in understanding the effects of ‘sub-concussive’ head impacts on health and performance.”

The authors addressed my comment about how many times pupil size was measured.  Given that it was measured once/eye pre- and post is there any documentation of the stability of pupil size over repeated trials.  In other words, how stable is pupil size at pre-participation?

Unfortunately we currently do not have any data regarding the stability of the pupil size prior activity in this cohort, but we are measuring this in the aforementioned follow up study that is now in recruitment to address this issue. We have also added the following statement to line 226-233 to highlight this as a key addition to any future studies:

“It is important to note, however, that whilst these variables did not show statistically relevant differences, the pre-post changes in these measurements may show clinically meaningful changes [43]. Understanding this would require measurements of pupil stability prior to sparring or physical activity to quantify the noise inherent in a dynamic system. This would also allow any changes to be appreciated within context. Future studies should, therefore, include multiple pre-sparring measures under controlled conditions to allow pupil stability and clinically meaningful changes to be determined.”

We have also added the following statement to the limitation section of the manuscript on lines 255-257

“Equally, the use of only one PLR measurement per eye prior to sparring meant that pupil stability of the cohort remains unknown. Future studies should therefore include a series of repeat measures prior to sparring and over several hours and days post sparring to determine…..”